# Arterial Thoracic Outlet Syndrome—A Case Study of a 23-Year-Old Female Patient Diagnosed Using a Thermal Imaging Camera

**DOI:** 10.3390/healthcare12171725

**Published:** 2024-08-29

**Authors:** Michał Żołnierczuk, Tomasz Skołozdrzy, Maciej Donotek, Zbigniew Szlosser, Piotr Prowans, Małgorzata Król, Bianka Opałka, Kamil Orczyk, Anna Surówka

**Affiliations:** 1Department of Vascular Surgery, General Surgery and Angiology, Pomeranian Medical University, 70-111 Szczecin, Poland; 2Department of Plastic, Endocrine and General Surgery, Pomeranian Medical University, 72-010 Szczecin, Poland; 3Department of Imaging Diagnostics and Interventional Radiology, Pomeranian Medical University, 72-010 Szczecin, Poland; 4Department of Histology and Developmental Biology, Faculty of Health Sciences, Pomeranian Medical University, 71-210 Szczecin, Poland

**Keywords:** thoracic outlet syndrome, thermal imaging, vascular surgery

## Abstract

We present the case of a 23-year-old woman who reported weakness in the left upper limb, decreased warmth, numbness in the fingers, pain in the clavicular region, and a severe cold sensation in the limb. A thermal imaging camera examination was performed for diagnostic purposes, which guided further diagnostic and therapeutic management towards arterial thoracic outlet syndrome (aTOS). Following surgery and rehabilitation procedures, significant remission of symptoms was achieved and the patient’s condition improved. This is the first report on the diagnosis of aTOS using thermal imaging, paving the way for further clinical research into this effective, rapid, and radiation-free method of diagnostic imaging. Conclusion: Thermal imaging is one of the most effective, readily available, and patient-safe methods for diagnosing vascular disease associated with flow disruption.

## 1. Introduction

Thoracic outlet syndrome (TOS) is a group of disease entities that are caused by the compression of the neurovascular bundle through surrounding structures of the upper limb [1]. The neurovascular bundle usually comprises the subclavian artery passing into the axillary artery and the axillary vein passing into the subclavian vein and the lower trunk of the brachial plexus. These structures exit the thorax, passing between the scalene muscles into the scalene triangle. They are bounded from above by the clavicle and from below by the first rib. They then pass the pectoralis minor muscle and run distally up to the upper limb, supplying it vascularly and nervously [2]. The space through which the vessels and nerves of the upper limb run is very limited, which sometimes is associated with the compression of bony or muscular structures on the elements of the neurovascular bundle, causing an impediment to proper vascular flow, as well as a limitation of the efficiency of nerve conduction [3].

Based on the compression of the structures of the neurovascular bundle, a distinction is made between neural TOS (nTOS), the most common, initially comprising up to 90–95% of TOS cases; venous TOS (vTOS), about 4% of cases; and arterial TOS (aTOS), 1% of cases [4,5]. The prevalence of this syndrome is estimated to be approximately 1–3/100,000, although it may be underestimated due to diagnostic difficulties resulting from clinicians’ misdiagnoses. Women aged 20–50 years are most commonly affected, and the etiology ranges from congenital anatomical defects resulting from excessive tightness of the structures surrounding the neurovascular bundle to acquired defects resulting from an accident or an inflammation involving the aforementioned structures [6].

Symptoms of TOS include upper limb paresthesia, which can be caused by both artery and nerve compression, sensory disturbances, neck pain, and quadriceps. Ischemia of the upper limb can also lead to rapid fatigue, weakness, and abnormal skin warming. TOS may manifest as swelling or tissue thickening in the fingers, resulting from venous insufficiency [5,7,8]. However, these symptoms pose many diagnostic difficulties, as the patient’s appearance may vary depending on the compressed structure. Currently, there is no standardized diagnostic procedure to distinguish between the different types of TOS, although a differential diagnosis of TOS can be made on the basis of the patient’s appearance and additional imaging studies with provocative tests [9].

Arterial TOS is diagnosed on the basis of characteristic symptoms presented by the patient, including pain, reduced body heat, weakness and fatigue in the upper limb, as well as on the basis of duplex ultrasound, plethysmography, pulse oximetry, computerized tomography angiography (angio-CT) images, and magnetic resonance angiography (angio-MR) images, using provocative maneuvers [5,9,10]. There is, however, a method that makes it possible to identify blood supply abnormalities in the upper limb and does not require the administration of contrast agents. This test uses a thermal imaging camera, comparing the blood supply to both upper limbs depending on their position. This article presents the case of a 23-year-old woman diagnosed with aTOS using a thermal imaging camera.

## 2. Case Report

In 2022, a nodular lesion and swelling involving the joint appeared in the left wrist area of a 23-year-old female. In addition, the patient noticed a slight cold sensation in her left hand, accompanied by numbness in her fingers. She had never experienced similar symptoms before since she had no chronic illnesses and no one in her family had autoimmune diseases. She was a physical worker and these symptoms began to hinder her daily work. In November 2022, she attended a surgeon who diagnosed her with ganglion and extensor muscle tendinitis. Her treatment included antibiotics and anti-inflammatory drugs. Rehabilitation treatments, including whirlpool massages, cryotherapy, lasers, joint mobilization, and ultrasounds, were prescribed. These treatments were unsuccessful, which was the reason for a visit to the surgical emergency room in March 2023. At that time, the patient presented increased numbness in the fingers of her left hand, decreased warmth, a strong feeling of cold, and pain in the left clavicle area. These symptoms were particularly exacerbated after physical exertion. On physical examination, a positive Phalen’s sign was noted, as well as a positive exercise test with elevated limbs. Laboratory tests showed a leukocytosis of more than 14,000/µL without any significant abnormalities. A chest X-ray and Doppler ultrasound without provocative maneuvers of the upper limb arteries were performed, but no significant pathology was detected. Due to the patient’s reported uncharacteristic symptoms and normal ultrasound findings, an additional thermal imaging camera test with a provocation test was performed to see if there would be a change in limb temperature when raising the hands above the head, during which the patient reports the greatest exacerbation of their symptoms. The patient was asked to raise her hands above her head so that the position of her hands exceeded a 90-degree angle, hold the position until the discomfort and numbness worsened, and then lower her hands to the starting position. The examination was carried out using a Flir 335 thermal imaging camera. A significant difference in temperature was observed between the left and right upper limb, as presented in Figure 1.

After a thermal imaging camera examination, where changes in the heat of the left upper limb starting above the elbow joint were visualized, it was decided to perform an angio-CT of the arteries of the upper limb with contrast. This examination was performed with the left upper limb elevated and bent at the elbow. At the level of the posterior slit of the inclined muscles, a stenosis of the left subclavian artery of approximately 70% in relation to the right subclavian artery was observed. The left axillary vein was also compressed between the subclavian artery and the pectoralis minor muscles. The vascular changes are presented in Figure 2. On examination, a diagnosis of aTOS was suspected.

Due to persistent symptoms of finger numbness and swelling of the wrist, a visit to a surgical clinic was recommended to rule out carpal tunnel syndrome and median nerve dysfunction. An ultrasound of the left wrist was performed in April 2023, which showed no joint effusion. From the deviations, a trace of fluid probably indicating residual ganglion was visualized. The flexor trochlea was not thickened, and the ligamentous apparatus was unchanged. In addition, magnetic resonance imaging of the left wrist showed a trace effusion in the region of the scaphoid bone. A normal median nerve without signs of oedema was observed. As part of the outpatient care, left upper limb arterial stenosis was confirmed using a Doppler ultrasound with a provocation test. We modelled the axillary artery by the inclined muscles with a subsequent narrowing of its lumen of more than 50% and a decrease in flow velocity values in the radial artery when examined using arm elevation. These symptoms were not observed at rest. In comparison, in the other upper limb, ultrasound showed no significant abnormalities.

After excluding median nerve dysfunction and other pathologies in the left hand, aTOS was diagnosed and the patient was qualified for surgery. Prior to surgery, an angio-CT was performed again to better visualize the site of stenosis. The surgical procedure took place in October 2023. An incision in the axillary fossa area under the edge of the pectoralis major muscle exposed the muscle, which was separated by lifting upwards; the pectoralis minor muscle and its upper edge were exposed. Above the muscle, the first rib was exposed and separated from the tissues subperiosteally. The rib was excised from a 6 cm long incision border. From a separate incision above the clavicle and along the sternocleidomastoid muscle, the oblique muscles and brachial plexus were exposed. A peripheral fragment of the anterior oblique muscle was excised. The intraoperative decompression of the subclavian vessels and the left brachial plexus was achieved. The patient felt well after the surgery. She was referred to a day rehabilitation center and a surgical clinic for a postoperative follow-up and suture removal. At the orthopedic clinic in November, it was decided to repeat the provocative examination with a thermal imaging camera. This examination showed no temperature difference between the left and right upper limb. The results are shown in Figure 3.

An angio-CT study was also performed, which showed no abnormalities in vascular blood flow during the provocative test. The result of this study is shown in Figure 4.

After the follow-up at the surgical clinic, the patient proceeded to receive rehabilitation for the limb. On admission to the rehabilitation center in December 2023, she reported weakness in her left upper limb, complaints of neck pain around the surgical scar, and anteroinferior thoracic pain on the left side. This pain was exacerbated by deep breathing and by turning her head to the left. In addition, she reported sensory disturbances in the operated area and discrete numbness in her left hand. Despite the complaints, the patient’s condition improved significantly compared to her preoperative state. Currently, the patient reports a slight weakness of the limb that does not impede daily functioning. Pain and minor numbness in the fingers persist when the patient physically works with her arms raised above her head for prolonged periods of time. In addition, she reports persistent slight sensory weakness in the surgical wound and in fingers I, II, and III. The cold sensation has subsided and the temperature of both limbs is comparable.

## 3. Discussion

The main symptoms of aTOS are arterial, but patients may also present with neurogenic or venous symptoms. Classic symptoms of aTOS syndrome include a cold limb, weakness, and pain due to distal ischemia. The difficulty in diagnosing aTOS stems from the fact that other disease entities present with similar symptoms, such as fibromuscular dysplasia, Buerger’s disease, diabetes mellitus, Takayasu’s, cryofibrinogenemia, or uremic arteriolopathy, and must be considered in the differential diagnosis [11]. In addition, symptoms resulting from upper limb peripheral nerve ischemia, as well as difficulties in venous blood outflow resulting from the co-occurrence of compression of the axillary and subclavian veins by surrounding structures, may be among the components of aTOS syndrome [9]. In the patient presented above, these symptoms were overlapping—decreased warmth and muscle weakness were due to impaired arterial blood flow. The numbness of the fingers and the sensory disturbances were probably due to ischemia of the median nerve, as the nerve structure itself and the ligamentous elements within the carpal canal were intact in the imaging studies performed. The initial swelling of the hand and wrist may have been due to impaired venous blood flow from the upper limb, as similar symptoms were not observed in the right upper limb and the patient had not suffered trauma or been diagnosed with rheumatological diseases.

The rarity of the disease, the atypical symptoms, and the lack of a standard clinical procedure to diagnose TOS means that the syndrome is often overlooked by clinicians. Even when the symptoms are indicative of TOS, the differential diagnosis between the TOS types concerned can often cause further diagnostic and treatment difficulties. However, some symptoms may indicate the relevant type. Increased symptoms of oedema and cyanosis may be more indicative of vTOS. nTOS manifests with increased weakness, numbness, as well as paresthesia and muscle atrophy. In addition to the typical symptoms listed above, aTOS may also manifest with Raynaud’s phenomenon, the ulceration of the fingers, and signs of peripheral embolism [9]. nTOS, the most common variant of TOS (90–95% of cases), should be considered first in the differential diagnosis [4,5]. However, when symptoms related to venous or arterial circulation failure occur, additional imaging studies such as angio-CT and Doppler ultrasound should be performed to check the patency of the upper limb vessels. Epidemiological data on disease prevalence, such as age and gender, can also be considered when differentiating TOS subtypes. Unlike neurogenic TOS, there is no gender preference in the arterial form, but some studies have shown a trend towards females [12]. The mean age of patients in the databases for aTOS ranges from 34.0 ± 15.4 years to 40.3 ± 2.2 years [13,14]. However, the age of the patient in our study was 23 years, which is below this range.

Despite the severity of symptoms and diagnostic difficulties, treatment of aTOS is effective and safe in most patients. Due to their young age, a small group of patients have concomitant diseases, such as atherosclerosis and diabetes, which could negatively affect the treatment process and the healing of the surgical wound. In addition, the procedure itself, which removes the cause of aTOS, which is a compression of the neurovascular bundle, has a high treatment success rate. Nineteen patients with aTOS were able to achieve complete healing and full rehabilitation after surgery. The researchers removed bony structures compressing the upper limb arterial vessels and created a subclavian-axillary bypass to restore arterial continuity. One patient required reintervention due to bypass occlusion, although this procedure was performed without further complications [13]. This study demonstrates the excellent efficacy of treating aTOS with the restoration of arterial continuity. As the patient treated at our center had symptoms of aTOS primarily during overhead elevation and with increased physical activity and, in addition, imaging studies did not show damage to the subclavian or axillary artery, it was decided to remove the bony structures without creating a subclavian-axillary bypass. As in the Marine et al. experiment, the patient’s symptoms became less intractable after the procedure.

Surgical treatment of TOS can be associated with many complications, such as damage to adjacent vessels, nerves, pneumothorax, and hemothorax [15]. For this reason, surgical treatment of TOS should be performed by well-qualified surgeons. Despite the surgical success, the patient from our center still had neurological symptoms after surgery. According to a study by Peek et al., who conducted a long-term evaluation of the treatment of a large group of patients with TOS, 54% of them achieved complete remission of symptoms within a year of the surgical procedure [16]. Other patients have extinguished paresthesia, pain in the upper limb, or the recurrence of symptoms that they experienced before surgery. The phenomenon of the persistence of neurological symptoms in the upper limb, despite a properly performed surgical procedure, may result from the difficulty of making a diagnosis and the length of time between the appearance of the first symptoms and surgery. The long period of nerve compression and abnormal blood supply to the peripheral nerves can lead to the persistence of neurological symptoms in the upper limb despite surgical success, as nerve fibrosis occurs over time [17]. To detect abnormal peripheral nerve function and assess the likelihood of the persistence of neurological symptoms after surgery in patients with TOS, electroneuromyography may be useful [18].

The case of the 23-year-old woman is probably the first time aTOS has been diagnosed using a thermal imaging camera. Duplex ultrasonography, contrast arteriography, and finger plethysmography have been used in the standard diagnostic management of the arterial variety of TOS. In some cases, CT/MR arteriography with provocative maneuvers was also used [9]. Unlike most types of radiological imaging, thermography offers a radiation-free and non-invasive method of diagnostic imaging and temperature measurement, which has been used at our center. In November 2021, the PROSPERO database found 39 ongoing review protocols with the keyword ‘thermography’. The reasons found for the use of a thermal imaging camera were Raynaud’s sign, diabetic foot, musculoskeletal injury, facial paralysis, spinal cord injury, and diabetic complications [19]. Thermography can therefore be a great tool in the diagnosis of vascular diseases, such as acute and chronic limb ischemia with limpid chroma, as well as small vessel pathology in autoimmune diseases or diabetes [20,21,22]. This method allows a quick and easily accessible examination of the patient, even within specialist clinics. The price of a thermal imaging camera ranges from 2500 to 25,000 euros, depending on the model, making it quite affordable. The thermal imaging camera also has the advantage of being handy, as all the equipment fits into a small suitcase. In addition, it can significantly reduce the time required to make a diagnosis, bypassing certain imaging studies, including MRI and Doppler ultrasound, or specialist consultations such as rheumatology, vascular surgeons, hand surgeons, and orthopedic consultations. There are not many publications on the use of this method in vascular pathologies, although the success of the use of this method in our diagnosis shows that thermography is an overlooked way of diagnosing vascular pathologies, and clinicians should consider its use as it opens up a new field of research into thermographic diagnosis.

## 4. Conclusions

Thermal imaging may be one of the most effective, readily available, and patient-safe methods for diagnosing arterial thoracic outlet syndrome. Its use may also be helpful in other diseases with impaired arterial flow along with other imaging studies used so far, while studies are needed to accurately determine the effectiveness of this method in selected disease entities.

## Figures and Tables

**Figure 1 healthcare-12-01725-f001:**
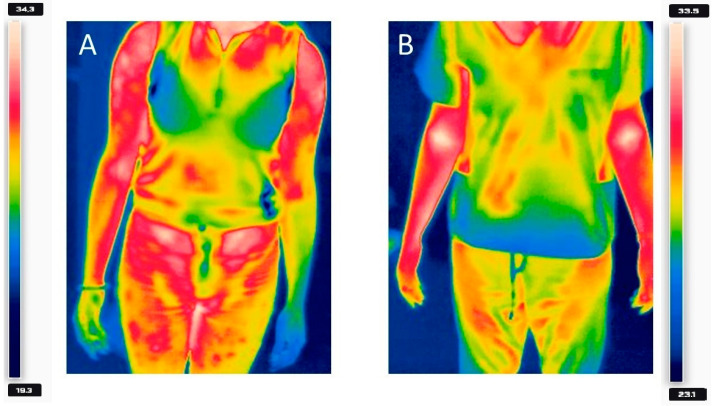
Comparison of changes in the upper limb warmth of the patient’s left upper limb after the provocation test (**A**) and with the physician (**B**). Temperature was given in degrees Celsius.

**Figure 2 healthcare-12-01725-f002:**
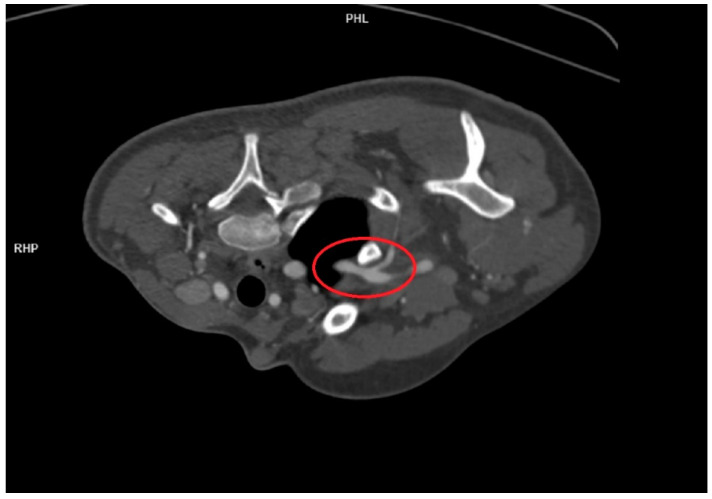
Angio-CT changes visualized in the patient before surgery. The location of the constriction of the left subclavian artery is marked with a red circle.

**Figure 3 healthcare-12-01725-f003:**
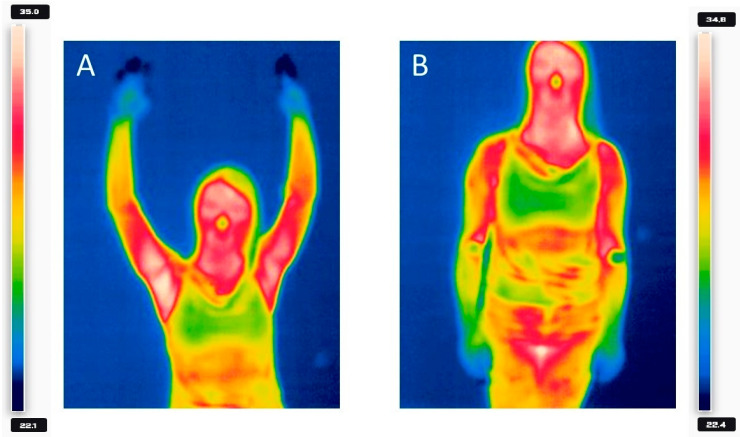
Comparison of the temperature of the patient’s right and left upper limb during the provocation test with raised (**A**) and lowered (**B**) hands. Temperature was given in degrees Celsius.

**Figure 4 healthcare-12-01725-f004:**
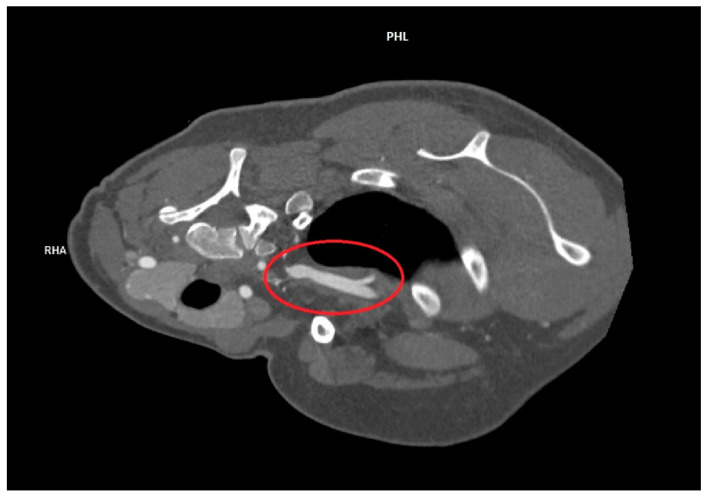
Angio-CT result after the procedure. The left subclavian artery is marked with a red circle.

## Data Availability

The data presented in this study are available upon reasonable request from the corresponding author.

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
