# Peer review of "Arterial Thoracic Outlet Syndrome—A Case Study of a 23-Year-Old Female Patient Diagnosed Using a Thermal Imaging Camera"

_healthcare, 2024, doi:10.3390/healthcare12171725_

Round 1

Reviewer 1 Report

Comments and Suggestions for Authors

I would like to thank the author for this original report on the use of a thermal imaging camera in the diagnosis of arterial thoracic outlet syndrome, a rare disease with ongoing debates regarding its diagnosis and exploration.

I would suggest several revisions throughout the manuscript. You need to clarify and refine certain points to improve your manuscript. The use of a thermal imaging camera should be nuanced in the abstract and conclusion, as it has only been tested on a single patient and further studies/reports are needed

Introduction:

- For more clarity, you should add the location "upper limb" in the first sentence (line 31-32)

- Regarding the distribution of neural, arterial, and venous TOS, (line 43-45) you should add 'initially' or 'historically' to the established percentages. There is a body of literature reporting different distributions/percentages depending on how arterial and venous compression are defined. The initial percentages are based on a definition that only considered complications such as aneurysm and thrombosis.

- In the description of the symptoms, and as reported in the literature, you can propose the example of paresthesia: it can be a symptom of neural compression but also arterial (as a sign of ischemia).

- line 63-64: you can add transuctaneous oxymetry recently reported

case report:

- line 87: please specify if the ultrasound was performed with or without provocative maneuvers.

- line 93: you should maybe specify the maneuver used, was the arm elevated at more than 90degree or was it a standard Roos/EAST maneuver?

- Figure 2: if posisble a reconstruction of the left subclavian artery will improve the clarity.

- line 127: please specify if the ultrasound assessment was negative/normal on the right arm.

- Figure 4: same comment as figure 3

- Figure 3: if I agree that both arm have now the same result on thermal imaging camera, the colour code is not the same than initially evaluate. I mean, in the second figure the patient seem to have,  as I can interpret colder cutaneous temperature, which could be due to sevreal factor (condition of assessment..). It could be great to add a scale to interpret these figures.

Discussion:

- You should discuss the fact that your patient continued to experience symptoms that look like neurological ones. While decompression of the artery can immediately improve arterial symptoms, chronic neural compression can lead to neural fibrosis. If decompression surgery is performed after several years of symptoms, it may not have a significant effect on these neurological symptoms. Electroneuromyography can provide valuable information when performed before and after surgery in these types of situations.

- While I agree with the effectiveness of the surgical treatment, I would nuance its safety. Indeed, rare but significant complications, mainly neurological, can occur, and the procedure requires a trained surgeon 

- It would be beneficial to provide more information on the camera system you used. Is it portable? Expensive? Has its intra/interoperator reliability been tested in other diseases or situations?

- You should nuance your conclusion and propose more applications of this assessment inTOS

Author Response

Dear Reviewer, 

Thank you very much for your valuable comments, which will certainly improve the merit of the work. I am sending below my responses to the posted comments.

Comment 1: "For more clarity, you should add the location "upper limb" in the first sentence (line 31-32)"

Response 1: The location of upper limb is now added to the main document

Comment 2: "Regarding the distribution of neural, arterial, and venous TOS, (line 43-45) you should add 'initially' or 'historically' to the established percentages. There is a body of literature reporting different distributions/percentages depending on how arterial and venous compression are defined. The initial percentages are based on a definition that only considered complications such as aneurysm and thrombosis."

Response 2: It has been improved in the text.

Comment 3: "In the description of the symptoms, and as reported in the literature, you can propose the example of paresthesia: it can be a symptom of neural compression but also arterial (as a sign of ischemia)."

Response 3: This aspect is now added.

Comment 4: "line 63-64: you can add transuctaneous oxymetry recently reported"

Response 4: Transcutaneous oxymetry has been added to the text.

Comment 5: "line 87: please specify if the ultrasound was performed with or without provocative maneuvers."

Response 5: This aspect is now specified in the text.

Comment 6: "Figure 2: if posisble a reconstruction of the left subclavian artery will improve the clarity."

Response 6: Unfortunately, due to the process of obtaining CT images and transferring the image to another device, the quality of the image decreased and despite many attempts, we were not able to obtain a clearer image. Given the anatomy of the lesion, this projection was the most appropriate and was chosen by the radiologist describing the lesion.

Comment 7: "line 127: please specify if the ultrasound assessment was negative/normal on the right arm."

Response 7: It now has been specified in the main document.

Comment 8: "Figure 4: same comment as figure 3"

Response 8: I presume that this comment was about figure 2, not figure 3. If so, the response to this comment is the same as "Response 6".

Comment 9: "Figure 3: if I agree that both arm have now the same result on thermal imaging camera, the colour code is not the same than initially evaluate. I mean, in the second figure the patient seem to have,  as I can interpret colder cutaneous temperature, which could be due to sevreal factor (condition of assessment..). It could be great to add a scale to interpret these figures."

Response 9: Temperature scale has been added to the figures.

Comment 10: "You should discuss the fact that your patient continued to experience symptoms that look like neurological ones. While decompression of the artery can immediately improve arterial symptoms, chronic neural compression can lead to neural fibrosis. If decompression surgery is performed after several years of symptoms, it may not have a significant effect on these neurological symptoms. Electroneuromyography can provide valuable information when performed before and after surgery in these types of situations."

Response 10: This part is now added to "discussion" section.

Comment 11: "While I agree with the effectiveness of the surgical treatment, I would nuance its safety. Indeed, rare but significant complications, mainly neurological, can occur, and the procedure requires a trained surgeon "

Response 11: This part is now added to "discussion" section.

Comment 12: "It would be beneficial to provide more information on the camera system you used. Is it portable? Expensive? Has its intra/interoperator reliability been tested in other diseases or situations?"

Response 12: Information on the name of the camera, the price range of similar devices, as well as their advantages were added to the work. In addition, information on which states the thermal imaging camera has been used in has been posted and supported by three new articles. The thermal imaging camera has been used in the pre- and post-operative evaluation of patients with lower extremity ischemia treated with stent grafts, in the evaluation of patients with diabetic foot, and in Reynaud's syndrome.

Comment 13: "You should nuance your conclusion and propose more applications of this assessment inTOS"

Response 13: Conclusion has been changed in a way that best describes the benefits of our study and paves the way for further research on thermal imaging

I hope that the amendments made meet your expectations and will significantly improve the merits of the work. 

Best regards,

Michał Żołnierczuk

Reviewer 2 Report

Comments and Suggestions for Authors

I congratulate the authors for a detailed and interesting case report. They have successfully applied the diagnosis and management of TOS due to arterial compression. They utilised thermal imaging camera in the diagnosis of the disease, which is a very interesting method. I suggest that the images in Figure 1 used in the article should be made in the same way as in Figure 2, i.e. in Figure 1, the images taken in a position with the arms raised upwards should be shared.

Author Response

Dear Reviewer, 

Thank you very much for your valuable opinion about our paper. 

We are aware that a photo of the patient taken during the provocative test with her hands raised above her head both before and after the procedure would have enriched the factual value of our work. Unfortunately, we must announce that during the trial, the photo with the hands raised above the head before the procedure was not recorded on the thermal imaging camera, but the evaluation was done in real time. For this reason, we are unable to attach the photo during the provocative trial. We hope that this inconvenience will not cause a great loss on our work, and that the remaining reviews will allow us to enrich the didactic aspect of the case study we presented.

Thank you again for your valuable comment and positive reference towards our publication.

Best regards,
Michał Żołnierczuk
Corresponding author

Reviewer 3 Report

Comments and Suggestions for Authors

The clinical case is very well described including all the diagnosis pathway and reelevant examinations made to the patient.

literature review bont in introduction and discussion well written and well documented with a good reading flow.

The authors highlight the utility of thermal camera and state in their conclusions:

"Thermal imaging is one of the effective, readily available and patient-safe methods  for diagnosing vascular disease associated with flow disruption" this is an overstatement given is a single case report

as this is their main key message Images should include comparatives from before and treatmenafter and a normal thermal image all in the same figure to be able to compare and also  provide  detailed description of the changes and clinical interpretation to give context of its use in clinical practice.

line 219 states:As in the Ma-219 rine et al. experiment, the patient's symptoms became less intractable after the proce-220 dure, and complete recovery and complete resolution of clinical symptoms would be 221 possible after undergoing a cycle of rehabilitation.

however, it is just an assumption that shouldn't be made as line 164 contradicts the complete resolution of syptoms:

Pain and minor numbness in the fingers persists when the patient 164 physically works with her arms raised above her head for prolonged periods of time. In 165 addition, she reports persistent slight sensory weakness in the surgical wound and in 166 fingers I, II and III. The cold sensation has subsided and the temperature on both limbs 167 is comparable.

Comments on the Quality of English Language

well written english

Author Response

Dear Reviewer, 

Thank you very much for your valuable comments, which will certainly improve the merit of the work. I am sending below my responses to the posted comments.

Comment 1: "Thermal imaging is one of the effective, readily available and patient-safe methods  for diagnosing vascular disease associated with flow disruption" this is an overstatement given is a single case report as this is their main key message Images should include comparatives from before and treatment after and a normal thermal image all in the same figure to be able to compare and also  provide  detailed description of the changes and clinical interpretation to give context of its use in clinical practice."

Response 1: The conclusion has been changed in a way that more accurately defines the intent of our work. We are aware that a photo of the patient taken during the provocative test with her hands raised above her head both before and after the procedure would have enriched the factual value of our work. Unfortunately, we must announce that during the trial, the photo with the hands raised above the head before the procedure was not recorded on the thermal imaging camera, but the evaluation was done in real time. For this reason, we are unable to attach the photo during the provocative trial. We hope that this inconvenience will not cause a great loss on our work, and that the remaining reviews will allow us to enrich the didactic aspect of the case study we presented.

Comment 2: "Line 219 states:As in the Ma-219 rine et al. experiment, the patient's symptoms became less intractable after the proce-220 dure, and complete recovery and complete resolution of clinical symptoms would be 221 possible after undergoing a cycle of rehabilitation.

however, it is just an assumption that shouldn't be made as line 164 contradicts the complete resolution of symptoms:

Pain and minor numbness in the fingers persists when the patient 164 physically works with her arms raised above her head for prolonged periods of time. In 165 addition, she reports persistent slight sensory weakness in the surgical wound and in 166 fingers I, II and III. The cold sensation has subsided and the temperature on both limbs 167 is comparable."

Response 2: A comment that speculated a complete recovery after a period of rehabilitation has been removed. In addition, a section was added to the discussion of what the patient's persistent symptoms may be due to despite the effectiveness of the treatment.

I hope that the responses on these valuable comments will greatly improve the value of the publication. In addition, thank you again for your valuable comments and positive reference towards our publication.

Best regards,
Michał Żołnierczuk
Corresponding author

Round 2

Reviewer 1 Report

Comments and Suggestions for Authors

Dear Authors,

I really appreciate your work in addressing the comments I initially made.

I only noticed one typo error: there is a dot before reference number 11.

I have no further comments on this new version.

Reviewer 3 Report

Comments and Suggestions for Authors

I thank the authors for their responses. I think the manuscript can be accepted on its current form

Comments on the Quality of English Language

none